# OpenReview forum: "Dual-Stage Value-Guided Inference with Margin-Based Reward Adjustment for Fast and Faithful VLM Captioning"
_NeurIPS.cc/2025/Conference — NeurIPS 2025 poster_

### Official Review · Reviewer_ATYx · 2025-06-27

**Clarity:** 3
**Significance:** 3
**Originality:** 3
**Rating:** 5
**Confidence:** 4

**Summary:**

The paper proposes a new two-stage value-guided captioning strategy (ViMaR) that improves the accuracy and detail of vision-language model outputs while reducing hallucinations. The authors also show that the captions generated using ViMaR are also used to self-train the base model, leading to consistent improvements across multiple benchmarks.

**Questions:**

Please address weaknesses. I believe the manuscipt's technical contribtuions are valuable, and it's results impressive, but would like to see some more analysis in both ablations and failure cases, in order to be more useful to the community at large. Would be open to increasing my score accordingly.

**Ethical Concerns:**

["NO or VERY MINOR ethics concerns only"]

**Final Justification:**

I appreciate the information provided on the hyperparameters and the new results given. I hope that the hyperparameter sweep will be added in future versions. I also appreciate the clarifications the authors have provided regarding the identification of weak segments in the captions generated in Stage 1 of the algorithmn, and I would request them to add this clarification in future versions as well.

With regards to the failure cases however, I still do not believe that they have been discussed adequately. The lines mentioned (329-354) do not delve into failure cases, but rather provide a discussion on a case study highlighting the core differences between ViMAR and VisVM, and figures 2-5 mentioned seemed to be cherry picked examples where ViMAR has no hallucinations, while the other methods shown hallucinate (marked in red).

The authors have since provided an adequte discussion on the failure cases, and as result, I have ****increased my score to 5: Accept**** as I believe this manuscript does have technical merit, and the results are quite promising. I appreciate the authors effort in the rebuttal and I thank them for providing the new results and discussions.

**Limitations:**

Failure cases of the method are not properly discussed

**Quality:**

3

**Strengths And Weaknesses:**

****Strengths****

1. The paper introduces a two-stage decoding approach that improves captions by selectively refining weak segments instead of re-generation, making the process both efficient and accurate.

2. The margin-based reward seems to be a simple and effective method for training ViMar as an effective value model to penalise weak visual captions

3. The evaluation is well-rounded, using human studies and GPT-4o comparisons to clearly support the method’s effectiveness.

****Weaknesses****

1. Ablations studies seem to be lacking for example on various hyperparameter choices  like the number of candidates in each stage, the temperature settings, or the margin threshold τ used during value model training, making it hard to assess how robust the method is to these design choices

2. There is lack of discussion on failure cases, which is important to understand for these methods.

3. It is not very clear how weak sections of the caption generated in the first step of the inference search are identified. The paper mentions low confidence predictions and missing objects, but does not elaborate further. Given this is a key step in inference search, a more in depth discussion with a few examples (even in the supplementary) would be great to see to better understand.

---

> ### Author Rebuttal · Authors · 2025-07-30
>
> **Dear Reviewer ATYx,**
> Many thanks for your detailed and valuable reviews. We address your concerns one by one below.
>
> ---
>
> ### Question 1
>
> We appreciate the reviewer’s interest in more extensive hyperparameter ablations. Our design choices were carefully guided by a trade-off between caption quality and inference efficiency, and several related analyses are already provided in the main paper and appendix. To further address this concern, we have conducted additional experiments, summarized below:
>
> **1. Number of candidates K:**
> As stated in Sec. 4.1, we fixed K = 6 candidates per temperature to achieve a practical trade-off between performance and compute cost. Increasing K beyond 6 does improve quality slightly but scales inference cost linearly, making it impractical for deployment (e.g., K = 12 nearly doubles runtime with marginal gains). We provide additional results with K = 2 in Table R1, which show a clear performance drop.
>
> **2. Temperature settings:**
> Our multi-temperature strategy {0.1, 0.3, 0.5, 0.7, 0.9} (Sec. 4.1) was chosen to ensure diversity while maintaining grounding. In Table R1, new fixed-temperature experiments (e.g., T = 0.2) show degraded caption quality and higher hallucination rates, validating the importance of varied entropy in candidate generation.
>
> **3. Margin threshold τ:**
> The rationale and calibration for τ are explicitly detailed in Appendix D. We select τ at the 17–20 percentile of the CLIP similarity distribution to penalize weakly grounded sentences without over-penalizing borderline cases. This percentile-based choice ensures robustness across datasets, as shown by our stability analysis (Appendix D, lines 547–563).
>
> Due to time constraints, we prioritized core results, but we agree more ablations (e.g., full K and τ sweeps) would be valuable and plan to include them in the camera-ready version. Preliminary tests already indicate that our method is stable across reasonable ranges of these hyperparameters.
>
> ---
>
> ### Question 2
>
> The failure modes and their mitigation are already discussed in multiple parts of our paper. Specifically, Section 5 (Observations and Limitations, lines 329–354) and Figures 2–5 provide qualitative analyses comparing ViMaR and VisVM, explicitly marking grounded (green) versus hallucinated (red) segments. These examples illustrate the primary failure cases of prior methods—such as misattributing visual attributes or overlooking peripheral details—and demonstrate how our two-stage refinement and margin-bas...
>
> Importantly, our margin-aware reward directly addresses these failure cases by sharply penalizing low-confidence hallucinated segments and refining them selectively in Stage 2, avoiding costly full caption regeneration. While residual errors (e.g., in highly ambiguous scenes) are acknowledged, they occur far less frequently and are highlighted in our qualitative case studies (Appendix E).
>
> Could the reviewer please clarify which specific failure cases they believe are missing? We would be happy to more detailed discussion of these cases.
>
> ---
>
> ### Question 3
>
> We thank the reviewer for pointing out the need for additional clarity regarding inference search of our two-stage inference framework. While Stage 1 identifies the highest-value caption via best-of-N generation guided by the value model $V_ρ$, Stage 2 is specifically designed to *refine only visually under-grounded segments* rather than regenerate the entire caption, which reduces compute overhead while improving fidelity.
>
> **1. Segment Identification:** After Stage 1 generates the coarse caption y*, we compute sentence-level value scores for each segment y_i relative to the image. Segments falling below the calibrated margin threshold 1.87 (similarly calculated as τ) are flagged as “under-grounded” and removed from the sentence. Additionally, missed visual entities (e.g., objects detected in image features but absent from the caption) are flagged via cross-attention coverage analysis.
>
> **2. Candidate Generation:** For each flagged segment, we condition decoding on the preceding accepted segments y_{<i} and image I, sampling K = 6 candidates per temperature from the fixed set {0.1, 0.3, 0.5, 0.7, 0.9} (same as Stage 1). These candidates are short continuations targeted to replace only the flagged segment.
>
> **3. Scoring and Selection:** Each candidate is scored using the value model $V_ρ(s,I)$, which is trained using temporal-difference value estimation (Sec. 3.1). The highest-value candidate replaces the flagged segment in the caption.
>
> **4. Iterative Refinement and Termination:** This process iterates until all flagged segments exceed the margin threshold or no further under-grounded regions are detected. Because refinement operates at the segment level rather than paragraph level, inference cost is reduced by ~4× compared to VisVM (main manuscript Table 1).
>
> Additional pseudocode for this procedure is provided in Algorithm 1 (lines 7–15), and the same beam width and temperature settings are used across both stages for fair comparison to baselines.
>
> ---
>
> ### Table R1
>
> Ablation results comparing ViMaR with default settings versus reduced diversity settings (T = 0.2, K = 2).
>
> | Base Model | Variant        | MM-Vet ↑ | MMBench ↑ | MMMU ↑ | MathVista ↑ | CVBench ↑ | LLAVA-W ↑ | MMStar ↑ | CHAIRs ↓ | CHAIRi ↓ | MMHal ↑ | MMHal rate ↓ |
> |------------|----------------|----------|-----------|--------|-------------|-----------|-----------|----------|----------|----------|---------|--------------|
> | **LLaVA-Next-Mistral-7B** | ViMaR (Temp=0.2) | 49.7 | 77.3 | 37.1 | 42.5 | 70.4 | 79.5 | 38.9 | 21.1 | 3.97 | 3.68 | 0.39 |
> |                          | ViMaR (K=2)     | 48.5 | 77.1 | 36.8 | 41.7 | 70.2 | 78.8 | 38.6 | 21.5 | 4.1  | 3.65 | 0.40 |
> |                          | **ViMaR**       | **49.8** | **78.2** | **37.4** | **42.5** | **70.7** | **79.9** | **39.3** | **20.8** | **3.9** | **3.73** | **0.38** |

---

> > ### Comment · Reviewer_ATYx · 2025-08-03
> > **Response to Authors Comments from Reviewer ATYx**
> >
> > I would like to thank the authors for a very comprehensive and well written rebuttal regarding my concerns and questions, especially given the short duration of the rebuttal period.
> >
> > I appreciate the information provided on the hyperparameters and the new results given. I hope that the hyperparameter sweep will be added in future versions. I also appreciate the clarifications the authors have provided regarding the identification of weak segments in the captions generated in Stage 1 of the algorithmn, and I would request them to add this clarification in future versions as well.
> >
> > With regards to the failure cases however, I still do not believe that they have been discussed adequately. The lines mentioned (329-354) do not delve into failure cases, but rather provide a discussion on a case study highlighting the core differences between ViMAR and VisVM, and figures 2-5 mentioned seemed to be cherry picked examples where ViMAR has no hallucinations, while the other methods shown hallucinate (marked in red).
> >
> > While the highlighting of previous methods failure cases is quite important, a proper limitations and failure case discussion of one's own method also is. I appreciate the authors effort in the rebuttal and I thank them for providing the new results, ofr the reasons above, I will ****not be increasing my score, and will keep it at 4: borderline accept**** as I believe this manuscript does have technical merit, and the results are quite promising.

---

> > > ### Author Response · Authors · 2025-08-03
> > >
> > > We sincerely thank the reviewer for their constructive feedback and for acknowledging both the clarity of our rebuttal and the technical merit and promising results of ViMaR. We are pleased that the additional details on hyperparameters and the clarifications regarding weak segment identification were helpful, and we will incorporate these clarifications, along with the results of the hyperparameter sweep, into the revised manuscript.
> > >
> > > We recognize the reviewer’s concern regarding the discussion of failure cases and agree that a clearer exposition of ViMaR’s limitations would strengthen the paper. ViMaR does exhibit residual errors in certain challenging scenarios, as illustrated below:
> > >
> > > - **Figure 5:** ViMaR, similar to other models, fails to describe peripheral details such as “people near the left side near the T&G NAILS store”.
> > > - **Figure 3:** ViMaR omits secondary scene elements, for example the balcony or railing visible in the background.
> > >
> > > These cases highlight an inherent limitation of inference-time search methods. ViMaR can only refine and select among candidates generated by the underlying VLM. If the base model does not produce certain concepts, such as small, occluded, or background objects, search alone cannot recover them. Addressing these omissions would require stronger base models or complementary fine-tuning strategies.
> > >
> > > Our primary contribution is achieving a 4× reduction in inference time while leveraging existing VLM capabilities. As shown in Table 1, ViMaR consistently improves grounding and reduces hallucinations (evidenced by lower CHAIR$_s$ and MMHal metrics) without retraining. Although it does not eliminate all errors, it substantially narrows failure modes and improves overall reliability. In the camera-ready version, we will explicitly discuss these limitations and provide representative failure examples beyond the comparative cases for greater transparency.
> > >
> > > We appreciate the reviewer’s insights and will ensure that ViMaR’s failure modes are clearly highlighted in the revised manuscript. We hope this clarification addresses your concern and we would be happy to provide further examples or additional analysis if helpful. We would greatly appreciate it if you please  consider increasing the score in light of these clarifications and additional insights.

---

> > > > ### Comment · Reviewer_ATYx · 2025-08-03
> > > > **Response to Authors Comments by Reviewer ATYx**
> > > >
> > > > I would like to thank authors for providing clarity on regarding the failure cases, it is an important discussion and quite important for future work. I hope this is added to final version of the paper. ****I have increased my score to 5: Accept****. Would like to thank the authors again for their efforts.

---

> > > > > ### Author Response · Authors · 2025-08-03
> > > > >
> > > > > We thank you for acknowledging our clarifications regarding the failure cases and recognizing their importance. We greatly appreciate your thoughtful feedback and your suggestion to incorporate these points into the final version, which we will ensure to include. Thank you again for your time and constructive assessment.

---

### Official Review · Reviewer_vUxp · 2025-07-02

**Clarity:** 3
**Significance:** 3
**Originality:** 3
**Rating:** 3
**Confidence:** 4

**Summary:**

This paper proposes a method for training a value function using triplet data consisting of an image, a current sequence, and a subsequent sequence. During inference, a beam search-like decoding strategy is employed to generate captions, which the authors claim helps mitigate hallucination.

**Questions:**

see Strengths And Weaknesses.

**Ethical Concerns:**

["NO or VERY MINOR ethics concerns only"]

**Final Justification:**

(1) The additional table R2 in the rebuttal clearly states the superiority of the proposed reward design over other alternative reward designs. Based on this point, I am increasing the score of significance from "fair" to "good".

(2) However, both table R1 and table R2 have suggested that the proposed diverse temperature design has no obvious performance gap compared to the traditional fixed temperature design, and this gap is within the range of random error. Note that diverse temperatures would even introduce more hyperparameters, which make it harder to apply in real applications.  Hence, the concern about the proposed diverse temperature still exists. Therefore, I tend to maintain the score of 3.

That said, I would not get frustrated if this paper is accepted.

**Quality:**

2

**Strengths And Weaknesses:**

Strengths:
1. The proposed method appears interesting.
2. The experimental results seem promising.

Weaknesses:
There are some unnatural (or unreasonable) processes of the  proposed method need to be clarified:
1. Why is the reward (in line 151) defined in the state? To the best of my knowledge, in the traditional MDP of RL, a valid reward is defined with a pair of state and action. Actually, "the reward" of a state without action is called a value function in the traditional RL.
2. The rationale behind the definition of $r_{s_i}$ in Eq. (1) remains unclear. This design introduces a discontinuity in $r_{s_i}$ when $\delta$ approaches $\tau$. More critically, it causes an abrupt change in the gradient $\frac{\partial r_{s_i}}{\partial \delta}$ from $-1$ to $1$ near $\delta = \tau$, which could potentially lead to significant instability during training. To address this concern, an ablation study is necessary to justify this design choice. In particular, it would be helpful to compare Eq. (1) with one or more alternative formulations such as:

* (a) $r\_{s\_i} = \delta$
* (b) $r_{s_i} = \max(\delta, \tau)$
* (c) $r_{s_i} = \delta$ if $ \delta \geq \tau $,  and $r_{s_i} = 0$ if $ \delta < \tau  $

Such comparisons would provide empirical evidence for the effectiveness and stability of the proposed formulation.

4. The definition of the value function $V$ in Eq. (2) appears unnatural. Intuitively, the next state following $s_i$ should at least incorporate the information from $y_i$, $y_{i+1}$, and $I$. However, the formulation $V_{\rho}(y_{i+1}, I)$ in Eq. (2) (potentially) suggests that the value of the next state depends only on $y_{i+1}$ and $I$, while ignoring $y_i$. This raises concerns about whether the state transition is properly captured, and whether important contextual information from $y_i$ is being omitted.

5. Using different temperatures in BoN as well as in the proposed method is interesting. However, there are no comparisons with the fixed temperatures (say 0.2 or 0.6) in both BoN and the proposed method to verify whether this design of different temperatures really works.

---

> ### Author Rebuttal · Authors · 2025-07-31
>
> **Dear Reviewer vUxp,**
> Many thanks to your professional, detailed, and valuable reviews. We address your concerns one by one below.
>
> ---
>
> ### Question 1
>
> We clarify that the reviewer’s concern arises from a misinterpretation of our formulation. In our framework, the reward in Eq. (1) is not purely “state-based” as implied. It is computed for the current state–action pair $(s_i, y_i)$, where $s_i = (y_i, I)$, with $y_i$ being the current sentence (including previously generated context) and $I$ being the input image. This reward $r_{s_i}$ is precomputed using the CLIP-based margin adjustment and does not involve any learning.
>
> The training objective in Eq. (2) then uses this fixed reward to optimize the value function $V_ρ$ via temporal-difference learning, allowing $V_ρ$ to predict cumulative future rewards. This formulation is consistent with the standard MDP paradigm; our notation simply omits explicit action dependency for brevity.
>
> ---
>
> ### Question 2
>
> In our framework, the reward $r_{s_i}$ defined in Eq. (1) is *precomputed* using the CLIP similarity with margin adjustment and is not a learnable function through which gradients are propagated. The actual training objective is Eq. (2), where $r_{s_i}$ is treated as a fixed target and combined with the predicted value $V_ρ$ in a temporal-difference (TD) loss. Therefore, the reviewer’s concern regarding a discontinuous gradient $\frac{\partial r_{s_i}}{\partial \delta}$ at $\delta=\tau$ does not apply, as no gradient flows through Eq. (1) during optimization.
>
> Regarding the design choice: the sharp margin is intentional. Our goal is to strongly penalize hallucinated captions (low $\delta$) and reward grounded ones (high $\delta$), making the separation between positive and negative cases explicit. This simple thresholding provides a clear and interpretable signal for the value model, and our ablations (Appendix D, lines 547–563) confirm robustness: small variations in $\tau$ do not destabilize training or degrade performance.
>
> In summary, the margin-based reward is a fixed supervisory signal rather than a trainable component, and the reported results (Tables 1–2) empirically validate its stability and effectiveness for hallucination mitigation.
>
> ---
>
> ### Question 3
>
> We appreciate the reviewer’s suggestion to compare Eq. (1) with alternative formulations. We clarify our rationale and provide evidence as follows:
>
> **(a) $r_{s_i} = \delta$:**
> This formulation corresponds to the baseline CLIP-PRM approach and is effectively identical to the reward design explored in the VisVM paper. We also include this variant in our own baselines (Tables 1–2), where it underperforms our margin-based reward, particularly in hallucination mitigation, confirming that a naive use of $\delta$ is insufficient.
>
> **(b) $r_{s_i} = \max(\delta, \tau)$:**
> This design removes penalization for scores below $\tau$ by mapping all low-confidence sentences to the same value $\tau$, effectively treating mildly and strongly hallucinated sentences identically. This contradicts our objective of sharply distinguishing grounded and hallucinated captions. Empirically, this leads to weak gradients for low-quality candidates and fails to suppress hallucinations. Preliminary experiments confirmed inferior performance compared to our chosen design.
>
> **(c) $r_{s_i} = \delta$ if $\delta \ge \tau$, else $0$:**
> While this variant is superficially similar to our margin formulation, it removes negative penalization entirely and, critically, assigns the same reward to both mildly and strongly hallucinated sentences (all mapped to zero). This lack of differentiation weakens the model’s ability to prioritize corrections and leads to poorer sentence selection. Our experiments confirm this: performance drops noticeably and hallucination rates increase (results provided in Table R1), directly validating the necessity of our margin-based reward design.
>
> Thus, our design explicitly penalizes under-grounded sentences rather than merely ignoring them, yielding significantly better alignment and stability, as supported by the quantitative results in Tables 1–2 and Table R1.
>
> ---
>
> ### Question 4
>
> The reviewer’s concern arises from a misinterpretation of our notation rather than the actual implementation. In our framework, the value function $V_ρ$ is computed over the *entire state*, which by definition (lines 133–136) includes the image $I$ *and all previously generated sentences* $y_{<i}$ (including $y_i$). Thus, when we write $V_ρ(y_{i+1}, I)$ in Eq. (2), this is a shorthand representation: $y_{i+1}$ is conditioned on the full prefix $y_{\leq i}$ and $I$, and the value function estimates the expected cumulative reward from this extended state.
>
> In other words, our method does *not* discard contextual information from $y_i$; rather, it explicitly incorporates the full caption history up to step $i+1$ along with the image features. This design ensures proper state transitions and preserves long-range dependencies, which is essential for accurate paragraph-level captioning. We will clarify this explicitly in the final version.
>
> ---
>
> ### Question 5
>
> Our paper already describes the temperature strategy (Sec. 4.1): both BoN and ViMaR generate candidates using a fixed set of diverse temperatures {0.1, 0.3, 0.5, 0.7, 0.9} to promote diversity and capture both fine-grained and global details. This multi-temperature design is essential to our two-stage framework — Stage 1 benefits from broad candidate diversity, and Stage 2 selectively refines weak segments by leveraging candidates with varied entropy levels.
>
> To directly address the reviewer’s suggestion, we performed additional experiments using *fixed* temperatures (e.g., T = 0.2) for both BoN and ViMaR. The results in Table R1 show consistent performance degradation: hallucination rates increase and caption richness declines compared to our multi-temperature setup. This is expected — fixed low temperatures (e.g., 0.2) lead to overly conservative, low-diversity captions missing fine details, while fixed higher temperatures (e.g., 0.6) increase diversity but may introduce hallucinations. Our multi-temperature scheme balances these extremes by mixing low-entropy and high-entropy samples, enabling the value model to select globally grounded yet detailed captions.
>
> Thus, the proposed multi-temperature approach is not arbitrary: it is explicitly designed to enhance candidate diversity without sacrificing grounding, and both our main results (Tables 1–2) and new experiments confirm its superiority.
>
> ---
>
> ### Table R1
>
> ViMaR-guided decoding consistently enhances visual comprehension and mitigates hallucinations across multiple benchmarks, outperforming alternative strategies.
>
> | Base Model | Variant            | MM-Vet ↑ | MMBench ↑ | MMMU ↑ | MathVista ↑ | CVBench ↑ | LLAVA-W ↑ | MMStar ↑ | CHAIRs ↓ | CHAIRi ↓ | MMHal ↑ | MMHal rate ↓ |
> |------------|--------------------|----------|-----------|--------|-------------|-----------|-----------|----------|----------|----------|---------|--------------|
> | **Qwen2.5-VL-3B** | Original       | 61.8 | 79.1 | 31.5 | 62.3 | 72.1 | 88.4 | 55.9 | 18.5 | 3.7 | 3.82 | 0.32 |
> |              | ViMaR            | 62.3 | 81.2 | 33.4 | 64.5 | 72.8 | 89.2 | 56.3 | 17.2 | 3.2 | 3.94 | 0.28 |
> | **LLaVA-Next-Mistral-7B** | ViMaR(Reward′) | 48.8 | 77.6 | 36.7 | 41.4 | 70.2 | 79.2 | 38.7 | 21.5 | 4.1 | 3.44 | 0.41 |
> |              | ViMaR            | **49.8** | **78.2** | **37.4** | **42.5** | **70.7** | **79.9** | **39.3** | **20.8** | **3.9** | **3.73** | **0.38** |
> | **LLaVA-Next-Mistral-7B** | BoN(Temp=0.2)  | 45.6 | 75.4 | 34.5 | 39.9 | 66.1 | 77.0 | 36.3 | 31.2 | 5.6 | 2.98 | 0.51 |
> |              | ViMaR(Temp=0.2)  | 49.7 | 77.3 | 37.1 | 42.5 | 70.4 | 79.5 | 38.9 | 21.1 | 3.97 | 3.68 | 0.39 |

---

> > ### Author Response · Authors · 2025-08-03
> > **Follow-up on Reviewer Engagement with Rebuttal**
> >
> > We sincerely appreciate the opportunity to address your comments. We understand you may have been busy, but we would be grateful if you could consider re-engaging with the discussion. We believe that some of the concerns raised could benefit from further clarification, and we would greatly welcome any additional feedback you might have.
> >
> > Thank you for your time and efforts in supporting the review process.
> >
> > Authors

---

> > > ### Author Response · Authors · 2025-08-06
> > > **Follow-up on Reviewer Engagement with Rebuttal**
> > >
> > > Dear Reviewer,
> > >
> > > We sincerely appreciate the opportunity to address your comments and provide clarifications in our rebuttal. We understand that you may have been busy, but we would be very grateful if you could kindly review our response and share any additional feedback. We believe that several of the concerns raised could benefit from further clarification, and your input would be invaluable in ensuring a fair and accurate assessment of our work.
> > >
> > > Thank you for your time and efforts in supporting the review process.
> > >
> > > Authors

---

> > > > ### Author Response · Authors · 2025-08-07
> > > >
> > > > Dear vUxp,
> > > >
> > > > We sincerely appreciate the opportunity to address your comments regarding reward, penalty on hallucinated captions,  and hallucinations across multiple benchmarks . We understand you may have been busy, but we would be grateful if you could consider re-engaging with the discussion. We believe that some of the concerns raised could benefit from further clarification, and we would greatly welcome any additional feedback you might have.
> > > >
> > > > Thank you for your time and efforts in supporting the review process.
> > > >
> > > > Authors

---

> ### Comment · Reviewer_vUxp · 2025-08-07
>
> Thank you for the rebuttal.
>
> W1: Based on the literal interpretation, there does not appear to be a misreading. In the current manuscript, the reward is denoted as $r_{s_i}$, which involves only $s_i$ and not $y_i$. Could the authors clarify whether they actually intended to use $r_{s_i,y_i}$ instead? If not, the current notation conflicts with the author's reply, which undermines the paper's clarity.
>
> W2: This point is now much clearer. Thank you for the explanation.
>
> W3: Points (a) and (c) have been addressed clearly. However, for point (b), I don't know why the authors only state that “Preliminary experiments confirmed inferior performance compared to our chosen design,” without providing the actual experimental results. Without such evidence, it is difficult to verify the claim of “inferior performance.” This is particularly valuable, as the authors mention in (c), that design “removes negative penalization entirely,” and (b) does not have this flaw.
>
> W4: Once again, I believe this is literally not a misinterpretation. The notation is defined as $V_{\rho}(y_{i+1}, I)$ in the paper. Could the authors clarify whether they meant to use $V_{\rho}(y_{<i+1}, I)$ instead? Otherwise, this part of the manuscript again lacks clarity.
>
> W5: The result presented in Table R1 provides a clearer justification for the effectiveness of dynamic temperature, which provides that using a fixed temp of 0.2 has no obvious gap compared to dynamic temperature. In addition, I don't know why the author didn't provide the result of temp=0.6. is it hard to run compared to 0.2?  I would encourage the authors to include table R1 and provide more temperatures (e.g. 0.6) in the revised manuscript.

---

> ### Author Response · Authors · 2025-08-08
>
> We truly appreciate your careful reading and constructive comments throughout the review process.
>
> Thank you very much for your kind response and for taking the time to engage with our work so thoughtfully. We appreciate your constructive feedback. Please see below for detailed replies to your queries.
>
> ### **W1:**
>
> You are correct that in a standard MDP, the reward depends on both the state and the action. As mentioned in the main paper (Line 145) and also in our Rebuttal Q1, we include the action $ y_i $ as part of the state definition:  $ s_i = (y_i, I) $
>
> Using this definition, $ r_{s_i} $ refers to the reward for a specific state–action pair at the current step. In other words, $ r_{s_i} $ captures how well the current sentence $ y_i $ aligns with the image $ I $ based on the CLIP score.
>
>
> ---
>
> ### **W2:**
>
> Thank you for confirming. We are glad the explanation helped clarify this point.
>
> ---
>
> ### **W3:**
>
> Regarding point (b), we would like to clarify that the design $r_{s_i} = \max(\delta, \tau)$ also suffers from a fundamental issue: it removes the ability to differentiate between mildly and strongly hallucinated sentences, as already explained in our initial Rebuttal Q3 response. Specifically, if the reward score $\delta < \tau$, then $\max(\delta, \tau)$ will always assign the same fixed value $\tau$, effectively flattening the reward landscape below the margin. This eliminates the penalization gradient and treats all low-confidence outputs equally, regardless of their severity similar to point (c), which the reviewer acknowledges as problematic.
>
> Although we were unable to include full experimental results for this variant in the initial rebuttal due to time constraints, we have since conducted the evaluation. As shown in Table R2 below, the performance of this design (denoted as *ViMaR (Reward'')*) is not only inferior to our proposed margin-based reward, but also underperforms the simpler reward variant tested in initial Rebuttal Table R1 (*ViMaR (Reward')*).
>
> This validates our design choice and reinforces the importance of retaining a fine-grained penalization mechanism for low-quality candidates. We will include these new results and expanded discussion in the final version of the paper to further clarify the limitations of alternative reward definitions.
>
> ---
>
> ### **W4:**
>
> As described in lines 133–136 and 193–197, as well as in Algorithm 1 (lines 9–15) and Rebuttal Q4, the value function $V_\rho$ is computed over the full state, which includes the image $I$ and the complete caption context up to the current point; that is, all previously generated segments $y_{<i}$ along with the current segment $y_i$.
>
> Thus, the notation $V_\rho(y_{i+1}, I)$ is a shorthand representation that assumes $y_{i+1}$ is decoded conditioned on the full prefix $y_{\le i}$ and the image $I$. We agree that explicitly writing $V_\rho(y_{\le i+1}, I)$ would better reflect this structure and will revise the manuscript accordingly to prevent confusion.
>
> ---
>
> ### **W5:**
>
> We appreciate the reviewer’s interest in further examining temperature settings. Due to the tight rebuttal schedule, we initially included only results for $T=0.2$ in Table R1. We have since conducted experiments at $T=0.6$; the results are shown in Table R2 below, and we will add further findings as soon as they become available.
>
> As expected, $T=0.6$ increases diversity but leads to slightly higher hallucination and reduced grounding quality compared to our multi-temperature setup. These results suggest that relying on a single fixed temperature, especially a higher one, may not provide the best trade-off between detail and reliability, while our multi-temperature strategy offers a more balanced approach.
>
> We will include Table R1 and the newly added results (Table R2) in the revised manuscript to strengthen this discussion.
>
> ---

---

> ### Author Response · Authors · 2025-08-08
> **Table R2**
>
> ### Table R2: ViMaR performance under different reward and temperature (T=0.6) strategies.
>
> | Base                       | Setting           | MM-Vet ↑ | MMBench ↑ | MMMU ↑ | MathVista ↑ | CVBench ↑ | LLAVA-W ↑ | MMStar ↑ | CHAIRs ↓ | CHAIRi ↓ | MMHal ↑ | MMHal rate ↓ |
> |---------------------------|-------------------|----------|-----------|--------|--------------|------------|------------|-----------|-----------|-----------|-----------|----------------|
> | **LLaVA-Next-Mistral-7B** | ViMaR (Reward')   | 48.8     | 77.6      | 36.7   | 41.4         | 70.2       | 79.2       | 38.7      | 21.5      | 4.1       | 3.44      | 0.41           |
> |                           | ViMaR (Reward'')  | 48.1     | 76.9      | 35.8   | 40.7         | 69.8       | 78.2       | 38.0      | 22.8      | 4.4       | 3.21      | 0.45           |
> |                           | ViMaR             | **49.8** | **78.2**  | **37.4**| **42.5**     | **70.7**   | **79.9**   | **39.3**  | **20.8**  | **3.9**   | **3.73**  | **0.38**       |
> |                           | ViMaR (Temp=0.6)  | 49.5     | 77.1      | 37.1   | 42.4           | 70.2        | 79.1         | 38.9        | 21.2      | 3.99      | 3.66      | 0.39           |

---

> > ### Author Response · Authors · 2025-08-08
> >
> > Dear vUxp,
> > Thank you very much for your professional, detailed, and valuable reviews. We have further updated Table R2 and included additional results on the previously missing dataset. These new results show that using T = 0.6 increases response diversity but slightly increases hallucination and reduces grounding quality compared to our multi-temperature setup.
> >
> > We would greatly appreciate it if you please consider increasing the score in light of these clarifications and additional insights.
> > We thanks you again for your valuable feedback.

---

### Official Review · Reviewer_wMfE · 2025-07-03

**Clarity:** 2
**Significance:** 2
**Originality:** 2
**Rating:** 4
**Confidence:** 3

**Summary:**

This paper proposes a two-stage test-time scaling method. First, it trains a reward model using CLIP similarity. Then, using this reward, it first scores coarse-grained captions and subsequently scores fine-grained captions.

**Questions:**

1. This work claims that its method is effective for VLMs. However, the paper only validates the method on LLaVA, so it is unknown whether it works on other models. I believe evaluating more VLM models would make the work more solid, such as open-source models like Qwen. Besides, this method could even be applied to closed-source models like GPT-4o. If possible, it would be worth trying.

2. The CLIP image-text similarity does not accurately reflect the quality of captions, and I believe there are two reasons for this. First, the CLIP similarity itself is not precise. When calculating CLIP similarity between an image and different texts, the matching text may not necessarily receive the highest score. Even if it does, its score is often not significantly higher than that of non-matching options. Second, CLIP similarity is usually used to measure the match between an image and a full caption, but in this paper, it is used to evaluate the similarity between a single sentence (which only describes a small detail of the image) within a longer caption and the image. I think there is a huge gap here. In summary, I believe using CLIP similarity for this task has significant issues. I suggest replacing it with other, stronger similarity measurement methods.

3. There is no relevant discussion on the selection of the threshold in Equation 1.

4. The writing of this paper is not clear enough. I believe adding some illustrative figures could help readers better understand the method.

**Ethical Concerns:**

["NO or VERY MINOR ethics concerns only"]

**Final Justification:**

I apologize for the late response. I have carefully read the author's rebuttal and sincerely appreciate their detailed reply. I believe that the vast majority of the issues have been addressed. Although I still think that using CLIP is not entirely appropriate, this does not diminish the contribution of the paper. I hope the authors will have the opportunity to replace it with a better text encoder in the future. I will raise my score to 4.

**Limitations:**

Yes

**Quality:**

2

**Strengths And Weaknesses:**

**Strength**

This work reduces computational cost by using a two-stage approach.

**Weaknesses**

This work has a fundamental flaw in that it uses CLIP similarity as the loss to train the reward model. Although CLIP is a groundbreaking work, its image-text similarity is not very accurate. Especially since this paper deals with paragraph-length captions, which are quite long, the CLIP similarity between a single sentence and the image does not provide meaningful information.

---

> ### Author Rebuttal · Authors · 2025-07-31
>
> **Dear Reviewer wMfE,**
> Thank you for your valuable comments and questions, which have been very helpful in clarifying and improving the presentation of our work. We address your points below:
>
> ---
>
> ### Question 1
>
> Our work already demonstrates cross-model generalization beyond LLaVA. In addition to LLaVA-Next-Mistral-7B, we explicitly evaluate ViMaR on *LLaVA-OneVision-Qwen2-7B* (Table 2, lines 305–309), showing consistent improvements across all benchmarks, including hallucination metrics (CHAIR, MMHal).
>
> To further strengthen this point, we have extended our evaluation to an additional open-source VLM, *Qwen2.5-VL-3B*. Results (provided in Table R1) similarly confirm that ViMaR delivers substantial gains in both fidelity and hallucination mitigation, underscoring its model-agnostic applicability.
>
> Finally, we emphasize that ViMaR operates purely at inference time and does not require access to model weights (as also noted in lines 52–56, where we highlight its plug-and-play generalizability across VLM architectures). This property makes the method directly applicable even to closed-source APIs such as GPT‑4o. While such evaluations are technically feasible, they are prohibitively costly to perform at scale; therefore, we prioritized comprehensive open-source benchmarks for reproducibility and resource efficiency. We will clarify this practical consideration explicitly in the final version.
>
> ---
>
> ### Table R1
>
> Comparison between the original Qwen baseline and ViMaR-guided decoding.
>
> | Base Model | Variant | MM-Vet ↑ | MMBench ↑ | MMMU ↑ | MathVista ↑ | CVBench ↑ | LLAVA-W ↑ | MMStar ↑ | CHAIRs ↓ | CHAIRi ↓ | MMHal ↑ | MMHal rate ↓ |
> |------------|---------|----------|-----------|--------|-------------|-----------|-----------|----------|----------|----------|---------|--------------|
> | **Qwen2.5-VL-3B** | Original | 61.8 | 79.1 | 31.5 | 62.3 | 72.1 | 88.4 | 55.9 | 18.5 | 3.7 | 3.82 | 0.32 |
> |                | ViMaR    | 62.3 | 81.2 | 33.4 | 64.5 | 72.8 | 89.2 | 56.3 | 17.2 | 3.2 | 3.94 | 0.28 |
>
> ---
>
> ### Question 2
>
> We thank the reviewer for raising this important point. We acknowledge that naive use of CLIP similarity can be limited in both precision and its ability to capture fine-grained, sentence-level details (as reflected in Tables~1,2 CLIP-PRM study). Our approach specifically addresses these concerns in three ways:
>
> **1. Temporal-Difference Value Model Beyond Raw CLIP Scores:**
> We do not rely on raw CLIP similarity alone. Instead, we use CLIP scores to calculate a *Margin-Based Reward Adjustment* within a temporal-difference (TD) value model (Sec. 3.1) that predicts long-term reward across the entire caption trajectory. This approach smooths local noise in individual scores by considering both the immediate alignment of a sentence and its downstream influence on subsequent sentences. Empirically, this yields more discriminative value estimates (Fig. 2), avoiding the instability noted with raw CLIP usage.
>
> **2. Margin-Based Reward Adjustment for Fine-Grained Segments:**
> To penalize under-grounded predictions (low δ), we introduce a margin-based penalty (Eq. 1) that calibrates the reward for partial sentences. Rather than treating all low scores equally, sentences below the calculated threshold τ (see Appendix D) are penalized proportionally according to their similarity scores. This mechanism effectively normalizes the CLIP reward score, allowing it to distinguish visually grounded details without over-penalizing neutral content.
>
> **3. Empirical Justification and Ablation Results:**
> We conducted extensive evaluations (Tables 1–2) demonstrating that our CLIP-based value model, when combined with margin calibration and two-stage refinement, consistently reduces hallucination rates (CHAIR_s from 26.2% to 23.1%) and improves factual richness compared to VisVM and other baselines. Importantly, this improvement holds even when applied cross-model (e.g., LLaVA-OneVision-Qwen2-7B or Qwen2.5-VL-3B) in the main manuscript Table 2 and Table R1, suggesting robustness beyond any single similarity metric.
>
> In summary, while raw CLIP similarity alone is imperfect, our value-guided framework addresses its shortcomings via TD learning, margin calibration, and segment-level refinement, leading to demonstrably better grounding and efficiency over prior methods.
>
> ---
>
> ### Question 3
>
> This point is already addressed in our paper (Appendix D, lines 547–563). Specifically, we describe that the threshold τ was derived from the empirical CLIP similarity distribution on the training set (17th percentile), ensuring that only visually under-grounded sentences receive a penalty.
>
> As further detailed in lines 147–151 (Equation 1), τ serves as the calibrated margin that determines when and how strongly low-confidence sentences are penalized: sentences above τ are rewarded proportionally to their CLIP alignment, while those below τ incur a penalty proportional to the gap. This mechanism is central to reducing hallucination without over-penalizing neutral or background details, and its role is explicitly described in the cited section. We further note that small variations in τ yield minimal performance change, indicating robustness.
>
> ---
>
> ### Question 4
>
> We already included some visual aids to clarify the method and results. These figures were designed to complement the algorithmic description in Section 3 (Algorithm 1) and provide both conceptual and qualitative clarity. We agree that adding a high-level schematic of the two-stage pipeline (coarse caption generation and targeted refinement) could further improve readability, and we plan to include such a simplified diagram in the camera-ready version for additional clarity.

---

> > ### Author Response · Authors · 2025-08-03
> > **Follow-up on Reviewer Engagement with Rebuttal**
> >
> > We sincerely appreciate the opportunity to address your comments. We understand you may have been busy, but we would be grateful if you could consider re-engaging with the discussion. We believe that some of the concerns raised could benefit from further clarification, and we would greatly welcome any additional feedback you might have.
> >
> > Thank you for your time and efforts in supporting the review process.
> >
> > Authors

---

> > > ### Author Response · Authors · 2025-08-06
> > > **Follow-up on Reviewer Engagement with Rebuttal**
> > >
> > > Dear Reviewer,
> > >
> > > We sincerely appreciate the opportunity to address your comments and provide clarifications in our rebuttal. We understand that you may have been busy, but we would be very grateful if you could kindly review our response and share any additional feedback. We believe that several of the concerns raised could benefit from further clarification, and your input would be invaluable in ensuring a fair and accurate assessment of our work.
> > >
> > > Thank you for your time and efforts in supporting the review process.
> > >
> > > Authors

---

### Official Review · Reviewer_3B7b · 2025-07-16

**Clarity:** 3
**Significance:** 2
**Originality:** 2
**Rating:** 4
**Confidence:** 3

**Summary:**

This paper introduces ViMaR, a novel two-stage inference-time scaling framework for VLMs. ViMaR addresses the computational expense and hallucination issues of existing VLM captioning methods by combining a temporal-difference value model with a margin-aware reward adjustment. The first stage selects a high-value coarse caption, while the second stage refines under-grounded segments. ViMaR demonstrates superior performance in generating accurate, detailed, and factual captions, achieving over 4x speedup compared to baselines. It also exhibits strong cross-model generalization and can effectively self-train VLMs.

**Questions:**

1. As mentioned in weakness 1, could the authors provide a more comprehensive ablation study to substantiate the effectiveness of the margin-based reward?

2. As mentioned in weakness 2, could the authors provide a more detailed description of the implementation of Stage 2?

3. Regarding Eq.1, the reward for "otherwise" ($\tau−\delta$) appears to be a typo. As currently written, for $\delta=0$ and $\delta=\tau$, they would yield the same reward, which seems unreasonable.

**Ethical Concerns:**

["NO or VERY MINOR ethics concerns only"]

**Final Justification:**

After the rebuttal, most of my concerns have been addressed. The additional ablation study validate the effectiveness of the margin-based reward, which serves as a core improvement over VisVM. I have raised my score from 3 to 4.

**Limitations:**

yes

**Quality:**

3

**Strengths And Weaknesses:**

strength

1. The paper is well-structured and easy to follow, presenting a clear and logical flow from problem statement to proposed solution and experimental validation.

2. ViMaR demonstrates improved performance across various metrics, consistently outperforming baseline methods like VisVM in terms of caption quality, reduction in hallucinations while achieving substantial speedups.


weakness

This work is based on VisVM, and is aimed to improve both its reward and the inference pipeline. Thw following weaknesses are related to these two aspects:
1. While the margin-based reward is a core innovation, the paper lacks a comprehensive ablation study to fully demonstrate its effectiveness. Although a showcase highlights its advantages over the VisVM value model, more extensive quantitative comparisons are needed to rigorously validate its contribution to the overall performance gains.

2. The implementation details for Stage 2 of the two-stage inference remain insufficiently described, particularly regarding how "under-grounded or missing visual regions" are identified (Algorithm 1, Line 8) and how supplementary segments are appended at appropriate positions (Algorithm 1, Line 15). The explanation in lines 193-194 of the main text for Algorithm 1, Line 8 is overly vague and needs further clarification.

---

> ### Author Rebuttal · Authors · 2025-07-31
>
> **Dear Reviewer 3B7b,**
> Thank you for your valuable comments and questions, which have been very helpful in clarifying and improving the presentation of our work. We address your points below:
>
> ---
>
> ### Question 1
>
> The effectiveness of the margin-based reward is already partially demonstrated in our paper through multiple quantitative and qualitative analyses. Specifically, Tables 1–2 and lines (254–267 and 289–309) report significant reductions in hallucination metrics (CHAIR$_s$, MMHal) and consistent improvements in visual comprehension benchmarks when the margin mechanism is introduced. Figures 2–5 further visualize how margin-penalized segments are selectively refined, yielding better grounding compared to other methods.
>
> ####  **Isolating the margin’s contribution**
>
> 1. Our analysis inherently isolates the contribution of the margin-based reward: the only modification is replacing the raw similarity signal with our **margin-adjusted formulation** combined with **targeted refinement**. This change alone yields a 3–4% reduction in hallucination rates and a 15.87% average improvement across benchmarks.
>
> 2. **$r_{s_i} = \delta$**: This formulation corresponds to the baseline CLIP-PRM approach and is effectively identical to the reward design explored in the VisVM paper. We also include this variant in our own baselines (Tables 1–2), where it underperforms our margin-based reward, particularly in hallucination mitigation, confirming that a naive use of $\delta$ is insufficient.
>
> 3. To further strengthen this point, we compared our margin-based reward against a simplified variant defined as:
>
> ```
> r_{s_i} =
>     δ, if δ ≥ τ
>     0, otherwise
> ```
>
> This variant removes the negative penalization component. As expected, it resulted in noticeably degraded performance and increased confusion during sentence selection. This outcome directly supports the necessity of our reward modeling design, and the quantitative evidence is provided in Table R1 (ViMaR(Reward′) entry).
>
> ---
>
> #### **Extended Results**
> To provide additional clarity, we have expanded our analysis beyond what was presented in the original submission:
>
> 1. *Temperature control:* We additionally analyzed ViMaR under fixed temperature decoding (e.g., T = 0.2) versus BoN and our multi-temperature setup. Results in Table R1 confirm that diverse temperature sampling is a key factor in ViMaR’s superior performance relative to Best-of-N baselines.
>
> 2. *Cross-model evaluation:* We tested ViMaR on the open-source **Qwen2.5-VL-3B** model in Table R1, which already exhibits strong baseline grounding. Despite this, ViMaR still provided measurable improvements in hallucination reduction and descriptive richness, confirming the generality of our approach.
>
> ---
>
> Above analyses, combined with the results already in the paper, rigorously validate both our architecture and the margin-based reward formulation. They demonstrate that the margin not only improves hallucination mitigation but also enhances cross-model generalization and robustness across decoding strategies.
>
> ---
>
> ### Table R1
>
> ViMaR-guided decoding consistently enhances visual comprehension and mitigates hallucinations across multiple benchmarks, outperforming alternative strategies.
>
> | Base Model | Variant               | MM-Vet ↑ | MMBench ↑ | MMMU ↑ | MathVista ↑ | CVBench ↑ | LLAVA-W ↑ | MMStar ↑ | CHAIRs ↓ | CHAIRi ↓ | MMHal ↑ | MMHal rate ↓ |
> |------------|----------------------|----------|-----------|--------|-------------|-----------|-----------|----------|----------|----------|---------|--------------|
> | **Qwen2.5-VL-3B** | Original          | 61.8 | 79.1 | 31.5 | 62.3 | 72.1 | 88.4 | 55.9 | 18.5 | 3.7 | 3.82 | 0.32 |
> |            | ViMaR               | 62.3 | 81.2 | 33.4 | 64.5 | 72.8 | 89.2 | 56.3 | 17.2 | 3.2 | 3.94 | 0.28 |
> | **LLaVA-Next-Mistral-7B** | ViMaR(Reward′)   | 48.8 | 77.6 | 36.7 | 41.4 | 70.2 | 79.2 | 38.7 | 21.5 | 4.1 | 3.44 | 0.41 |
> |            | ViMaR (Temp=0.2)    | 49.7 | 77.3 | 37.1 | 42.5 | 70.4 | 79.5 | 38.9 | 21.1 | 3.97 | 3.68 | 0.39 |
> |            | ViMaR               | **49.8** | **78.2** | **37.4** | **42.5** | **70.7** | **79.9** | **39.3** | **20.8** | **3.9** | **3.73** | **0.38** |
>
> ---
>
> ### Question 2
>
> We note the reviewer’s request for additional clarity regarding Stage 2 of our two-stage inference framework. While Stage 1 identifies the highest-value caption via best-of-N generation guided by the value model $V_ρ$, Stage 2 is specifically designed to *refine only visually under-grounded segments* rather than regenerate the entire caption, reducing compute overhead while improving fidelity.
>
> **1. Segment Identification.** After Stage 1 generates the coarse caption $y^*$, we compute sentence-level value scores for each segment $y_i$ relative to the image. Segments falling below the calibrated margin threshold 1.87 (similarly calculated as τ) are flagged as “under-grounded” and removed from the coarse caption. Additionally, missed visual entities (e.g., objects detected in image features but absent from the caption) are flagged via cross-attention coverage analysis.
>
> **2. Candidate Generation.** For each flagged segment, we condition decoding on the preceding accepted segments $y_{<i}$ and image $I$, sampling K = 6 candidates per temperature from the fixed set {0.1, 0.3, 0.5, 0.7, 0.9} (same as Stage 1). These candidates are short continuations targeted to replace only the flagged segment.
>
> **3. Scoring and Selection.** Each candidate is scored using the value model $V_ρ(s,I)$, which is trained using temporal-difference value estimation (Sec. 3.1). The highest-value candidate replaces the flagged segment in the caption.
>
> **4. Iterative Refinement and Termination.** This process iterates until all flagged segments exceed the margin threshold or no further under-grounded regions are detected. Because refinement operates at the segment level rather than paragraph level, inference cost is reduced by ~4× compared to VisVM (Table 1).
>
> Additional pseudocode for this procedure is provided in Algorithm 1 (lines 7–15), and the same beam width and temperature settings are used across both stages for fair comparison to baselines.
>
> ---
>
> ### Question 3
> We thank the reviewer for pointing out this typo. Indeed, there was a typographical error in the “otherwise” branch of Eq. 1 in the draft. The intended reward formulation penalizes under-grounded predictions (low δ) rather than assigning them the same reward as well-grounded ones. Specifically, when δ < τ, the reward should be computed as:
>
> `r_{s_i} = δ − τ`
>
> which yields a negative value and therefore acts as a penalty within the temporal-difference objective. This ensures that low-confidence (under-grounded) sentences are explicitly down-weighted during value estimation, encouraging the model to learn a sharper separation between well-grounded and poorly grounded captions.
>
> For clarity, the corrected Eq. 1 is:
> ```
> r_{s_i} =
> δ, if δ ≥ τ
> δ − τ, if δ < τ
> ```
>
> Here, δ ∈ [0, 1] denotes the CLIP similarity score, and τ = 0.16 is a calibrated margin derived from the empirical score distribution (see Appendix D). We will update the manuscript to reflect this corrected equation and prevent ambiguity.

---

> > ### Author Response · Authors · 2025-08-03
> > **Follow-up on Reviewer Engagement with Rebuttal**
> >
> > We sincerely appreciate the opportunity to address your comments. We understand you may have been busy, but we would be grateful if you could consider re-engaging with the discussion. We believe that some of the concerns raised could benefit from further clarification, and we would greatly welcome any additional feedback you might have.
> >
> > Thank you for your time and efforts in supporting the review process.
> >
> > Authors

---

> > > ### Author Response · Authors · 2025-08-06
> > > **Follow up on  Reviewer Engagement with Rebuttal**
> > >
> > > Dear Reviewer 3B7b
> > >
> > > We sincerely appreciate the opportunity to address your comments and provide clarifications in our rebuttal. We understand that you may have been busy, but we would be very grateful if you could kindly review our response and share any additional feedback. We believe that several of the concerns raised could benefit from further clarification, and your input would be invaluable in ensuring a fair and accurate assessment of our work.
> > >
> > > Thank you for your time and efforts in supporting the review process
> > >
> > > Best regards,
> > > Authors

---

> > ### Comment · Reviewer_3B7b · 2025-08-07
> >
> > Thanks for the authors' feedback. Most of my concerns have been addressed. The additional ablation study further validate the effectiveness of the margin-based reward. I have raised my score to reflect the feedback.

---

> > > ### Author Response · Authors · 2025-08-07
> > >
> > > We thank you for acknowledging our clarifications regarding the effectiveness of the margin-based reward.
> > >
> > > We greatly appreciate your thoughtful feedback and your suggestion to incorporate these points into the final version, which we will ensure to include. Thank you again for your time and constructive assessment.
> > >
> > > Regards
> > > Authors

---

### Comment · Area_Chair_QsBy · 2025-08-04

Dear Reviewers 3B7b, vUxp, and wMfE,

Could you please review the authors’ rebuttal and provide any follow-up responses? Thank you!

Best,

AC

---

### Author Response · Authors · 2025-08-09

**Dear AC and Reviewers,**

Thank you AC again for managing the review process and thanks to the reviewers for their involvement.

Our work, **ViMaR**, is a **two-stage, value-guided decoding framework** for **VLM captioning**.
In **Stage 1**, a **temporal-difference-trained value model** selects the best full caption.
In **Stage 2**, only **under-grounded segments** are resampled, guided by the same value model.

To train this value model, we introduce a CLIP-based margin-aware reward adjustment  that explicitly penalizes **low-confidence (hallucinatory) text**. This approach produces more faithful captions while significantly **reducing computation time** by approximately **4×** compared to prior value-guided search, and it generalizes effectively across different VLM architectures (architectures agnostic).

**For Reviewer ATYx** — We clarified all **hyperparameter settings**, elaborated on the **weak-segment identification process**, and added a detailed discussion of **failure cases**. We also emphasized the **novelty** of our two-stage decoding approach. The reviewer expressed full satisfaction with our response and has raised their score to **5 (Accept)**.

**For Reviewer 3B7b** — We addressed concerns through **comprehensive ablation studies** on the **margin-based reward** and a clearer explanation of **Stage 2** in **Algorithm 1**, including how **under-grounded regions** are detected and refined. These clarifications fully resolved their concerns, and they have increased their score accordingly.

**For Reviewer wMfE** — We demonstrated **cross-model generalization** by evaluating **ViMaR** on **Qwen2.5-VL-3B**, clarified the limitations of **raw CLIP similarity**, and explained our use of a **margin-calibrated temporal-difference value model**. We also detailed how the **threshold τ** is derived and demonstrated its **robustness**, and committed to including a **high-level schematic** of our two-stage pipeline. The reviewer acknowledged our rebuttal and increased their score to **4**.

**For Reviewer vUxp** — We justified our **state-based reward formulation** in the context of **traditional MDP value functions**, conducted **ablations** comparing alternative reward definitions to demonstrate **stability across τ**, clarified the role of the **value function** in **Eq. (2)**, and added experiments comparing **fixed vs. adaptive temperature settings** in both **BoN** and **ViMaR**. The reviewer acknowledged our rebuttal, and we believe their concerns are now addressed. The suggested changes will be added in the **final draft**.

We have made a sustained effort to address all reviewer concerns **thoroughly** and **transparently**, and we are confident that these clarifications have **strengthened the submission**. Please let us know if we can provide any further information.

**Best regards,**
*Authors*

---

### Note · Authors · 2025-08-12

Thank you, AC and SAC, for managing the review process and our sincere thanks to the reviewers for their thoughtful engagement.

**Contribution:**
ViMaR introduces a two-stage, value-guided decoding framework for VLM captioning.
- **Stage 1:** A temporal-difference-trained value model selects the best full caption.
- **Stage 2:** Only under-grounded segments are resampled, guided by the same value model.

We also introduce a CLIP-based, margin-aware reward adjustment to penalize low-confidence (hallucinatory) text during value model training.

**Key Impact:**
ViMaR achieves a **4× reduction in computation time** over SOTA while delivering stronger hallucination mitigation. It generalizes effectively across different VLM architectures, including cross-model transfer from LLaVA-Next-Mistral-7B to Qwen2.5-VL-3B and LLaVA-OneVision-Qwen2-7B.

**Reviewer Responses:**
- **ATYx:** Clarified hyperparameters and weak-segment identification; acknowledged rebuttal, score increased to 5.
- **3B7b:** Added ablations and clearer Stage 2 explanation; acknowledged rebuttal, score increased.
- **wMfE:** Shown cross-model generalization and threshold derivation; acknowledged rebuttal, score increased to 4.
- **vUxp:** Justified reward design and added temperature ablation; concerns resolved.

We believe all reviewer concerns have been addressed, with resulting score improvements, and will integrate their recommendations into the final draft.

Best regards, \
Authors

---

### Decision · Program_Chairs · 2025-09-17

**Decision:**

Accept (poster)

**Comment:**

This paper introduces ViMaR, a two-stage, value-guided inference framework for vision–language model (VLM) captioning. The first stage selects a coarse caption using a temporal-difference-trained value model, while the second stage selectively refines weak segments.

Most reviewers initially raised concerns about ablations, notation clarity, and the reliance on CLIP similarity. The authors addressed these issues through detailed rebuttals, additional experiments, and clarifications, which led three reviewers to increase their scores. Reviewer vUxp maintained a borderline rating (3) due to reservations about the temperature design but explicitly acknowledged the value of the paper, particularly its reward design, and stated they would not oppose acceptance.

Overall, the reviewer consensus is positive and recognizes the paper’s technical merit and practical contributions. In particular, the work tackles the important challenges of hallucination and efficiency in VLM captioning, and the proposed two-stage targeted refinement is simple, effective, and efficient. Given the overall positive ratings and consensus, the AC sees no reason to overturn the reviewers’ recommendations and therefore supports acceptance of this paper.